# Structural–Functional Relationship of the Ribonucleolytic Activity of aIF5A from *Sulfolobus solfataricus*

**DOI:** 10.3390/biom12101432

**Published:** 2022-10-06

**Authors:** Alice Romagnoli, Paolo Moretti, Mattia D’Agostino, Jesmina Rexha, Nunzio Perta, Astra Piccinini, Daniele Di Marino, Francesco Spinozzi, Anna La Teana

**Affiliations:** 1Department of Life and Environmental Sciences, Polytechnic University of Marche, Via Brecce Bianche, 60131 Ancona, Italy; 2New York-Marche Structural Biology Center (Ny-MaSBiC), Polytechnic University of Marche, Via Brecce Bianche, 60131 Ancona, Italy

**Keywords:** aIF5A, Archaea, *Sulfolobus solfataricus*, RNA metabolism, SAXS, ribonuclease

## Abstract

The translation factor IF5A is a highly conserved protein playing a well-recognized and well-characterized role in protein synthesis; nevertheless, some of its features as well as its abundance in the cell suggest that it may perform additional functions related to RNA metabolism. Here, we have undertaken a structural and functional characterization of aIF5A from the crenarchaeal *Sulfolobus solfataricus* model organism. We confirm the association of aIF5A with several RNA molecules in vivo and demonstrate that the protein is endowed with a ribonuclease activity which is specific for long and structured RNA. By means of biochemical and structural approaches we show that aIF5A can exist in both monomeric and dimeric conformations and the monomer formation is favored by the association with RNA. Finally, modelling of the three-dimensional structure of *S. solfataricus* aIF5A shows an extended positively charged surface which may explain its strong tendency to associate to RNA in vivo.

## 1. Introduction

The translation factor IF5A belongs to the group of universally conserved factors participating in the process of protein synthesis [1,2]. It exists in Bacteria as EF-P, in Archaea as aIF5A and in Eukarya as eIF5A, three proteins characterized by a specific degree of sequence similarity (i.e., 35% aa identity between Eukarya and Archaea) but, above all, by a high degree of structural homology [3]. In addition, they share a common characteristic: the three proteins undergo a post-translational modification on the same conserved amino acid which is located in a region well characterized from a structural and functional point of view. The above mentioned modification is called hypusination in Archaea and Eukarya, whereas in Bacteria different types of modification have been described: lysinilation, 5-aminopentanolation and rhamnosylation [4,5]. Hypusine conversion occurs in all eukaryotes and requires two enzymes: a deoxyhypusine synthase (DHS), which catalyzes the formation of the intermediate deoxyhypusine by transferring the aminobutyl group of spermidine on the specific lysine (Lys 50 in human), and a deoxyhypusine hydroxylase (DOHH), which completes the reaction and leads to the formation of hypusinated-eIF5A [6]. Concerning Archaea, several species (*S. acidocaldarius*, *S. solfataricus*, *H. cutirubrum*, *T. acidophilum*) contain hypusine, some anaerobic species undergo only the first reaction, so they contain deoxyhypusine, while some other contain both variants of the protein [7,8]. Conversely, EF-P modification follows completely different pathways [9,10]. Despite a different chemical transformation and two different enzymatic pathways, the modified amino acid fulfills the same function: it aids peptide bond synthesis between amino acids which act as poor substrates for the peptidyl transfer reaction, and which would thus cause the stalling of the ribosome. This has been clearly demonstrated in Eukarya and Bacteria for poly-Proline stretches [11,12,13] and only hypothesized in Archaea [14]. Successive studies have shown that its function in translation elongation is not limited to the polyproline stretches but extends to other peptide contexts [15] and that eIF5A can be considered a more general translation factor also involved in translation initiation [16,17] and termination [15,18]. In addition to its well-established function in translation, several reports have proposed additional roles for a/eIF5A also in consideration of its great abundance [14,19] and many pieces of evidence point to a participation in RNA metabolism. Its structure suggests an RNA-binding capacity and, in fact, the human protein is endowed with a hypusine-dependent sequence-specific RNA-binding activity [20,21]. Other works have provided indications of an involvement of eIF5A in mRNA decay [22,23,24], and a direct RNA-degradation activity has been demonstrated in Archaea [14,25]. Wagner et al. [25] showed, for the first time, that aIF5A from *Halobacterium* sp. NRC-1 exhibits RNA cleavage activity, regardless of post-translational modification. They identified the specific charged amino acid residues responsible for the cleaving activity which occurred preferentially between adenine and cytosine in single stranded regions of RNA. The halobacterial aIF5A was further shown to bind to RNAs in vitro, but in contrast to the RNase activity, the hypusine residue was required to stabilize RNA-protein complexes [25]. Bassani et al. [8] investigated for the first time the activities of this protein in *S. solfataricus*, confirming the evolutionary conserved role of aIF5A as a translation factor, but also suggesting the hypothesis of a multitasking protein involved in RNA metabolism. Concerning the three-dimensional organization of the protein, a comparison among the crystallographic structures of eukaryotic and archaeal proteins available in databases (PDB IDs: *H. sapiens* 3CFP, *L. mexicana* 1XTD, *L. braziliensis* 1X6O, *S. cerevisiae* 3ER0, *P. horikoshii* 1IZ6, *M. jannaschii* 1EIF/2EIF and *P. aerophilum* 1BKB) indicates that both eIF5A and aIF5A are organized in two distinct domains predominantly composed of β-strands. The basic N-terminal domain is folded in a SH3-like barrel, found in other proteins related to translation and includes the region with highest conservation: the sequence surrounding the hypusination site. On the other hand, the acidic C-terminal domain harbors an OB-fold (oligonucleotide-binding fold), a five-stranded β-barrel known to bind nucleic acids similar to cold-shock domain and typical of other translational machinery components, such as eIF1A, eIF2α and of some ribosomal proteins [26]. In contrast, EF-P is organized in three domains [27]. The N-terminal domain shows a strong similarity to the N-terminal domain of a/eIF5A, while the other two are similar between them and to the C-terminal domain of a/eIF5A [3]. The protein has been described to act on the ribosome in a monomeric conformation [28,29]; nevertheless, some results indicate that, in particular conditions, it may undergo dimerization, for example, the X-ray analysis of the archaeal *M. jannaschii* aIF5A shows two crystal forms: monomer and dimer [30]. Other works, using biochemical and biophysical assays in yeast [26,31] and in Archaea [25], have shown that a/eIF5A can exist in both dimeric and monomeric forms and that the presence of RNA molecules is crucial to form and preserve the dimeric conformation. The presence of a dimeric organization as well as the possible involvement of RNA molecules opens up to a series of new hypothesis about additional functions of IF5A requiring further analyses.

Here, we have addressed some of the open questions concerning IF5A using the protein from the model Crenarchaeota *S. solfataricus* which has been extensively characterized in our laboratory [8,14,32]. We have used biochemical and biophysical approaches to investigate the structural organization of aIF5A in relation to its ability to bind and degrade RNAs. We demonstrate that the protein is associated with RNA molecules in vivo, and this association may promote the monomeric state of the protein. We confirm that aIF5A exerts an RNA-degrading activity on different types of RNA most likely mediated by the presence of complex structural elements. Small-angle X-ray scattering (SAXS) analysis show that the protein appears to be a monomer in solution and at high temperature and confirm that the presence of RNA molecules may stabilize the monomeric conformation. In summary, we provide new structural and functional insights supporting the moonlight functional profile of aIF5A protein related to RNA metabolism.

## 2. Materials and Methods

### 2.1. Cell Growth and S100 Lysate Preparation

*S. solfataricus* P2 strain was grown at 75 °C in Brock’s medium supplemented with 0.2% NZ amine and 0.2% sucrose, until OD600 of culture reached 0.6–0.8. Two liters of cells were pelleted and resuspended in 10 mL of lysis buffer containing the following: Tris-HCl pH 7.4 50 mM; NaCl 100 mM; Phenylmethylsulfonyl fluoride (PMSF) 1 mM; Dithiothreitol (DTT) 1 mM; MgAcetate 10 mM; Triton X-100 0.1%. After sonication, cell extract was clarified by centrifugation at 26,000× *g* for 30 min at 4 °C. The supernatant was subjected to ultracentrifugation at 100,000× *g* for 1 h at 4 °C, then supernatant was collected to obtain S100 extract [33,34].

### 2.2. Immunoprecipitation of Native aIF5A from S. solfataricus Lysate and RNA Extraction

The cytoplasmatic fraction of *S. solfataricus* lysate S100 was used for immunoprecipitation of aIF5A. An aliquot of S100 corresponding to 80 mg of proteins was incubated with 80 μL of anti-aIF5A antibody [8] or with pre-immune serum (80 μL), as negative control, and incubated on a rotating wheel at 4 °C for 2 h to allow specific binding of native aIF5A to its antibody. Each sample was then mixed with 50 μL of pre-equilibrated Protein G Dynabeads (Thermo Fisher Scientific, Waltham, MA, USA) in equilibration/wash buffer containing Tris-HCl pH 7.4 50 mM; NaCl 100 mM; MgAcetate 10 mM, Triton X-100 0.1%. Samples were incubated by rocking overnight at 4 °C and then placed on the magnetic device to remove unbound proteins. Beads were washed 3 times with buffer on the magnet, and aIF5A protein elution was obtained with 100 μL of Laemmli SDS sample buffer in incubation for 10 min at 95 °C. Bound fractions were collected and an aliquot of each fraction (input, unbound, last wash, bound) was kept aside for Western blot analysis performed using anti-aIF5A antibody as in [14]. The rest of the eluates were subjected to phenol/chloroform/isoamyl alcohol for RNA extraction and precipitated by ethanol. RNA was resuspended in DEPC-water and treated with a Turbo DNase RNase-free (Thermo Fisher Scientific, Waltham, MA, USA) digestion to remove genomic DNA from RNA samples. A PCR using 16S rRNA-specific primers (Appendix A) was performed to confirm the absence of DNA contamination. The RNA was quantified using Nanodrop 2000c spectrophotometer device and the quality was assessed by 8 M urea/8% polyacrylamide gel electrophoresis.

### 2.3. Reverse Transcriptase-PCR

The total RNA, extracted either from IP-aIF5A and IP-pre immune serum, was retrotranscribed using SuperScript™ III Reverse Transcriptase kit (Invitrogen, Waltham, MA, USA). Then, 50 ng of RNA was mixed with 1 µL random hexamer primers (100 ng/µL) and 1 µL of 10 mM dNTP mix in DEPC-H_2_O and incubated at 65 °C for 5 min, followed by 1 min on ice. Then, to the mixture were added: 5X first strand reaction buffer (Invitrogen cDNA synthesis kit), 0.1 M DTT, 1 µL of RNase inhibitor (40 U/µL) and 1 µL SuperScript III RT and DEPC-H_2_O to the final volume of 20 µL. The mixture was incubated for 5 min at 25 °C, then for 90 min at 50 °C. The RT was inactivated at 70 °C for 15 min. The resulting cDNA (2 µL) was PCR amplified with Phusion High-Fidelity PCR Master Mix (Thermo Fisher Scientific, Waltham, MA, USA) for 30 s at 98 °C, 30 s at 55 °C, and 45 s at 72 °C (30 cycles) with a final extension step of 10 min at 72 °C. Forward and reverse primers for 2508sh, 2184, 0910 mRNA and ncRNA 98, that amplified an internal region of about 200 bp of cDNA sequences, are listed in Appendix A. The corresponding PCR products were 230 nt (mRNA 2184), 225 nt (mRNA 0910), 110 nt (ncRNA 98) and 253 nt (mRNA 2508sh) in length. To confirm total DNA depletion of RNA samples, the PCR reactions were performed with the same amount of RNA as used for RT-PCR reaction without reverse transcriptase. PCR products were loaded into 6% polyacrylamide native gel. The electrophoresis was carried out at 10 mA for 40 min, using Tris-Borate-EDTA (TBE) as running buffer. Genomic DNA was used as positive control and H_2_O for negative control.

### 2.4. Density-Based Separation of Native aIF5A by 5–15% Glycerol Gradient Centrifugation

Cell pellets were resuspended in Extraction buffer (20 mM Tris-HCl pH 7.4; 10 mM MgAcetate; 40 mM NH_4_Cl; 1m M DTT) and lysed in the presence of allumina (2 g/g pellet) and 0.01 mg DNase (Sigma-Aldrich, Saint Louis, MS, USA). Lysates were centrifuged at 26,000× *g* for 30 min at 4 °C. Cleared extracts (0.5 mg) were loaded onto 12 mL continuous 5–15% glycerol gradients in sterile Extraction buffer, with and without pre-incubation with 0.05 mg RNase A (Sigma-Aldrich) for 30 min at 37 °C, and then centrifuged at 36,000 rpm for 17 h at 4 °C (rotor model SW40Ti; Beckman Coulter, Brea, CA, USA); fractions of 0.5 mL were collected using an AKTA FPLC machine. Collected fractions were precipitated with trichloroacetic acid (final concentration 10%) for 1h. Samples were then centrifuged at 15,000× *g* for 30 min at 4 °C, washed with 0.5 mL of cold acetone and centrifuged again under the same conditions. Proteins were resuspended in SDS sample buffer (64 mM Tris-HCl pH 6.8, 15% 2-mercaptoethanol, 10% glycerol, 2% SDS, 0.1% bromophenol blue) and used for Western blotting analysis. The experiment was repeated using a molecular mass standard (Gel Filtration Standard #1511901, Biorad, Hercules, CA, USA).

### 2.5. Recombinant aIF5A Purification

Recombinant *S. solfataricus* P2 aIF5A (ORF SSO0970) has been overexpressed either in *E. coli* ROSETTA (DE3)/pLysS (pETM11-N-His-aIF5A) and directly in *S. solfataricus* PH1–16 (pMJ05-aIF5A-C-His) and the recombinant proteins have been purified by affinity chromatography as previously described [8].

### 2.6. S. solfataricus Total rRNA and tRNA Extraction

Total RNA was extracted from *S. solfataricus* P2 pellet using Nucleospin RNA kit (Macherey-Nagel, Düren, Germany) and eluted with DEPC-H_2_O. tRNA was extracted from S100 lysate after phenol/chloroform treatment and precipitated by adding 2.5 vol of 95% ethanol [34,35]. Even though this preparation comprises a variety of cellular low-molecular-weight RNAs, the majority of it is tRNA, making it suitable to our applications.

### 2.7. rRNA 23S and 16S Isolation and Degradation Assays

*S. solfataricus* rRNA 23S was extracted from *S. solfataricus* 50S ribosomes and 16S from 30S ribosomes after phenol/chloroform treatment and precipitated by adding 2.5 vol of 95% ethanol and resuspended in DEPC-H_2_O. *S. solfataricus* ribosomes preparation was performed as described in Londei et al. [33]. Briefly, during the preparation of S100, the resulting pellet obtained from the ultracentrifugation is resuspended in buffer containing Tris-HCl pH 7.4 20 mM; MgAcetate 10 mM; NH_4_Cl 500 mM; DTT 2 mM and the high-salt-washed ribosomes are centrifuged on 10–30% sucrose gradients, obtaining the separation of the 50 and 30S subunits. The RNA-degradation assays were performed with 100 and 200 pmol of *S. solfataricus* N-His-aIF5A produced in *E. coli*, pre-activated for 10 min at 65 °C. RNA samples (1 µg of total rRNA, 23S or 16S rRNA and 2 µg of total tRNA for each sample) were incubated for 5 min at 85 °C. N-His-aIF5A, RNAs and buffer (10 mM HEPES pH 8.0, 100 mM KCl, 5 mM MgCl_2_, 5 mM β- mercaptoethanol, 5% glycerol) were mixed together and incubated at 65 °C for 30 min. As a control of RNA stability, a control of different type of RNAs without incubation and one without protein were added. The processes were stopped by adding RNA loading dye and incubated again for 10 min at 65 °C. The samples were placed onto 1.5% agarose gel and stained with Xpert Green DNA stain.

### 2.8. In Vitro Transcription and RNase Activity Assay of aIF5A

*S. solfataricus* 2184/0910 mRNAs and 98 ncRNA were in vitro transcribed as described above [14,36]. Primers are listed in Appendix A. RNA-degradation assays were performed as in [14].

### 2.9. Small-Angle X-ray Scattering (SAXS)

SAXS experiments have been carried out at the Diamond Light Source (Harwell Science & Innovation Campus, Didcot, UK) synchrotron, in the SAXS beamline B21, operating at wavelength λ = 1.000 Å and with a sample to detector distance 4.014 m. A first series of SAXS curves has been obtained from recombinant N-His-aIF5A in 20 mM Tris HCl (pH 7.7) and 2 *v*/*v*% glycerol, at five protein concentrations (0.5, 1, 2, 5 and 10 mg/mL) and with two different KCl amounts (60 and 1000 mM). Data have been recorded in the capillary heater at step of 5 °C from 20 °C to 63 °C, the maximum temperature achievable by the apparatus. The range of the scattering vector modulus q=4πsinθ/λ (2θ being the scattering angle) has ranged from 0.01 to 0.37 ${Å}^{-1}$. A second series of SAXS curves has been directly recorded at 63 °C on five samples of recombinant N-His-aIF5A (in the same buffer of the first series with 60 mM KCl) mixed with rRNA. The protein concentration of the four samples was 0, 0.5, 1, 4 and 6 mg/mL, respectively, whereas the concentration of rRNA on each of them was 2.5, 2.5, 2.5, 1.75 and 1 mg/mL. Buffer scattering has been recorded before each sample and the buffer contribution, corrected by transmission, has been subtracted from the SAXS pattern of each sample. Data in absolute scale have been obtained by using pure water as calibrant. To reduce the presence of protein aggregates, before SAXS measurements all protein samples have been centrifuged for 10 min at 14,000× *g* and stored at 4 °C.

### 2.10. SAXS Data Analysis

The experimental SAXS curves were represented in log–log panels, as Guinier plots and as Kratky plots. These representations made it possible to highlight the presence of large aggregates and smaller particles. A first analysis of the data was subsequently carried out by simultaneously adopting Guinier’s law and Porod’s law. Details are shown in the Appendix A. This analysis allowed the determination of the gyration radius $R_{g}$ of the smallest particles (Appendix A) and their apparent number of aggregation ($N_{agg,sml\;}$), Appendix A). Subsequently a more in-depth analysis was conducted using the GENFIT software and adopting the method described in detail in the Appendix A. This analysis was carried out with a simultaneous fit (global-fit) of all the experimental curves and considering the entire range of q (Appendix A). It should be noted that 10 conformations, representative of the 5A monomer, identified in a semi-qualitative way with the QUAFIT method, were adopted. The relative weight of these conformations was estimated by means of a simple thermodynamic model, whose parameters, with their dependence on KCl concentration, were obtained from the fit of the set of all curves (Appendix A).

### 2.11. Computational Analysis

We used the SwissModel web server to build the three-dimensional (3D) SM-Sso-IF5A and SM-Hsa-IF5A structural models [37]. The *S. solfataricus* aIF5A (UniProt ID Q97ZE8) structure was generated using the aIF5A from *Pyrobaculum aerophilum* as template (PDB ID 1BKB), whereas the *Halobacterium* sp. NRC-1 eIF5A (UniProt ID Q9HP78) structure was built employing the aIF5A from *Pyrococcus horikoshii* as template (PDB ID 1IZ6). Using ColabFold software [38] and the aminoacidic sequence, we predicted the *S. solfataricus* aIF5A Alphafold2 structure (i.e., AF-Sso-IF5A). The functional and structural analyses by means of the domains and functional sites classification on the aIF5A sequence from *S. solfataricus* (UniProt ID Q97ZE8) was performed using InterPro tool (access on 8 July 2022) [39]. We utilized the EMBOSS needle tool from the European Bioinformatics Institute (EBI online service) to perform pairwise alignment of the *S. solfataricus* and *H. NRC-1* IF5A sequences [40]. COBALT, a constraint-based alignment tool for multiple protein sequences provided in the NCBI website, was used for the multiple sequence alignment [41]. We employed PDB2PQR tool to convert the coordinates file from PDB to PQR and, to compute the electrostatic potential surface, we used PARSE forcefield. The APBS program was then used to solve the continuous electrostatic equations and obtain the electrostatic potential data [42]. We used ChimeraX to integrate the electrostatic potential into each protein’s 3D molecular surface for both eukaryotic and archaea IF5A proteins [43]. Then, we superimposed each structure on the SM-Sso-IF5A model to obtain the same spatial orientation.

## 3. Results

### 3.1. A Specific Subset of RNA Molecules Is Associated with S. solfataricus aIF5A In Vivo

aIF5A is an abundant protein in the cell and just a minor fraction of it is associated with ribosomal particles engaged in protein synthesis [14], suggesting that it may have other roles besides the one as translation factor. Starting from this concept, we first decided to evaluate the expression level of the protein in *S. solfataricus* cells. We measured, by Western blot analysis, the amount of aIF5A present in different phases of cell growth: early exponential (0.2 OD600), mid exponential (0.4 OD600), late exponential (0.8–1.3 OD600), and stationary phase (1.8 OD600) (Figure 1A). As shown in Figure 1B similar levels of aIF5A protein were detected for up to 34 h of growth (1.82 OD600), suggesting that the factor is expressed constitutively, and it is not subjected to regulation during growth. Different works in literature have presented IF5A as an RNA-binding protein in eukaryotes [20,21], while in some archaeal species, aIF5A can bind and also cleave RNA molecules [14,25]. In a deep sequencing analysis of RNA co-purified with recombinant aIF5A-C-His [14], a list of RNAs associated with the recombinant protein was identified, among them several mRNAs, some tRNAs and ncRNAs. However, those experiments were performed during purification of the recombinant protein at a temperature of 4 °C, far from the physiological one (75–80 °C). Therefore, in order to confirm the association of RNA molecules with the native protein in vivo, the three more representative RNAs were selected (based on the highest number of reads in aIF5A-C-His eluate compared to the mock control), two mRNAs and one ncRNA, and used for immunoprecipitation assays. The selected mRNAs are 2184 and 0910. The 2184 gene is 1184 nt long and encodes for a cell division control 6/orc1 protein homolog (cdc6-3); 0910 gene is 780 nt long and encodes for a putative cell division protein (KEGG database). Finally, ncRNA98 codes for a 90 nt long non-coding RNA whose function is unknown. To test if these RNAs are associated with native protein in vivo, RNA immunoprecipitation (RIP) was performed. Native aIF5A was immunoprecipitated from *S. solfataricus* post-ribosomal fraction (S100) using anti-aIF5A antibodies and magnetic Protein G Dynabeads. As a control of specific aIF5A-binding a pre-immune serum was used. Figure 1C shows the Western blot analysis of different aIF5A-immunoprecipitation steps. Figure 1D (lane 3) highlights the presence of the native aIF5A only in the sample incubated with anti-aIF5A, and the absence of the protein in the control sample incubated with pre-immune serum (Figure 1D, lane 4). RNAs directly bound to native aIF5A were then extracted from both this elution fractions, to ensure that the extracted RNAs were directly related to aIF5A and not to non-specific interactions with the magnetic beads, and DNase treated. The RNAs were retrotranscribed and cDNAs were used as a template for PCR amplification using specific primers that amplify internal sequences of mRNA 2184, mRNA 0910, nc98 and 2508sh mRNA, another *S. solfataricus* mRNA but not on a list of RNA co-purified with aIF5A-C-His [14]. Figure 1D shows that there is an amplification band only in PCR-reactions with oligonucleotides that amplified mRNA 2184, mRNA 0910 and ncRNA 98, chosen from the list of RNAs co-eluted with aIF5A and this confirms that this three RNAs are associated with the aIF5A protein also in vivo.

Early studies by Gentz et al. [31] and Dias et al. [26] suggested that yeast eIF5A exists as a dimer in solution, and the dimerization is RNA dependent. Starting from this observation, we wanted to analyze the effect of endogenous RNAs molecules on the structural organization of the native protein. Fractionation of the whole-cell lysate was performed through a 5–15% glycerol gradient and the localization of *S. solfataricus* aIF5A was monitored by Western blot analysis, in the presence (normal condition) or in the absence (after RNase A treatment) of its RNAs substrate (Figure 1E). In the control (Figure 1E, upper panel) *S. solfataricus* aIF5A is exclusively present in the low-molecular-weight fractions, consistent with the molecular weight of the monomer of the protein (17 kDa). Besides this, no other band corresponding to the formation of aggregates or high-molecular-weight protein complexes is visible under normal conditions. However, when the cell lysate is treated with RNase A, the protein localization shifts towards higher molecular weight fractions, and is detected around 35–40 kDa, which could indicate the formation of dimers or of complexes with other proteins (Figure 1E, lower panel).

### 3.2. S. solfataricus aIF5A Exerts Its Ribonucleolytic Activity on Long and Structured RNA

In addition to the ability to bind RNA, aIF5A was shown to possess an endoribonucleolytic activity being able to cleave in vitro transcribed mRNAs [14,25]. In order to further clarify this activity of the protein, we performed in vitro degradation assays using different classes of RNAs. In a first series of experiments, we selected two RNA types among the pool of aIF5A-interacting RNAs: mRNA and ncRNA. For this purpose, the three more representative RNAs identified previously (mRNA 2184, mRNA 0910, ncRNA 98), were in vitro transcribed and used in an in vitro RNA cleavage assay similarly to what we have already shown in a previous work [14]. Here, we have investigated further, performing a time-course experiment and comparing the activity of recombinant unmodified protein, purified from *E. coli* (N-His-aIF5A), with the recombinant hypusinated aIF5A, purified from *S. solfataricus* (aIF5A-C-His). Each of the two proteins was incubated at 65 °C for 25 min in the presence of the different length RNA substrates: mRNA 2184 (1184 nt), mRNA 0910 (780 nt), ncRNA 98 (90 nt), in an RNA/protein ratio 1:4. Every 5 min, an aliquot was withdrawn and analyzed by denaturing Urea-PAGE. Experiments were repeated at least three times and representative results are shown in Figure 2A–C and Appendix A. These results represent a clear indication that both unmodified and hypusinated aIF5A degrade the two mRNA substrates (2184, 0910) (Figure 2A,B), and that in both cases the presence of hypusine is irrelevant although in the case of mRNA 0910 the hypusinated protein shows a slightly more efficient cleavage activity (Figure 2B). In contrast the small ncRNA 98 is not degraded by aIF5A (Figure 2C and Appendix A). For each reaction a control sample of RNA without protein was incubated for 25 min at 65 °C, to prove the stability of these RNAs at high temperature. Taken together, these results indicate that aIF5A exerts its cleavage activity on the long mRNAs but not on the short ncRNA suggesting that the cleavage occurs on RNA which fold in complex secondary and tertiary structures.

To confirm this finding, we decided to test aIF5A activity on other RNA types: we selected, as examples of long and structured RNAs, 23S and 16S rRNA extracted from *S. solfataricus* ribosomal subunits, which were not present in the pool of aIF5A binding partners, and as an example of short RNA, the RNA mixture extracted from the post-ribosomal supernatant containing mainly tRNAs, some of which were found to bind aIF5A [14]. We incubated recombinant N-His-aIF5A either with a mixture of total rRNA or with the isolated 23S and 16S rRNA (from 50S and 30S ribosomal subunits, respectively), for 30 min at 65 °C and reaction mixtures were loaded on agarose gel. Results, presented in Figure 2D (lanes 1–4) and 2E show that both types of rRNA are degraded by aIF5A confirming the hypothesis that long and structured RNAs are a substrate for aIF5A activity. In both experiments, controls of RNAs alone with and without incubation at 65 °C were added to confirm the specificity of aIF5A-based degradation. The specificity of aIF5A activity was further confirmed by a zymogram assay which was performed, according to [14], using the endogenous protein purified from *S. solfataricus* and total rRNA included into the gel matrix (Appendix A).

Then, the mixture of small RNAs, which mostly includes tRNAs, was incubated with the protein at 65 °C for 30 min, and the results show that no cleavage can be observed for this type of RNA (Figure 2D, lanes 5–8).

### 3.3. Structural Characterization of S. solfataricus aIF5A in Solution by SAXS

We have demonstrated so far that aIF5A binds several types of RNAs and is able to degrade some of them, particularly those folding in complex secondary structure. In addition, some of our results indicate that the protein changes its conformation depending on its RNA-binding state, going from a monomer to a higher molecular weight complex upon RNase digestion. In order to further explore these features, we have used different structural approaches.

First, with the aim to obtain a detailed description of the three-dimensional (3D) organization of the protein and since no structures for *S. solfataricus* IF5A are available in the PDB database, we built a 3D model using the SwissModel online web server [37]. The IF5A structure from *Pyrobaculum aerophilum* (PDB ID 1BKB) was exploited as a template to generate the *S. solfataricus* aIF5A model (i.e., SM-Sso-IF5A). We selected *P. aerophilum* IF5A because of the significant amino acid sequence similarity, which results in a C-α RMSD of 0.120 Å between the two structures (Figure 3A). The structure of SM-Sso-IF5A appears to be similar to the eukaryal and archaeal counterpart [44]. This is not unexpected, given that SM-Sso-IF5A is a homology-derived structure. *S. solfataricus* aIF5A is mostly made of β-strands and comprises two different domains connected by a hinge linker (between β6 and β7) (Figure 3B). The N-terminus of *S. solfataricus* aIF5A contains six antiparallel β-strands (numbered from β1 to β6), with strands β1 and β6 hydrogen linked and strands β2-β3-β4-β5 forming a β-sheet. Between β1 and β2, there is a brief α-helix turn (VGE, residues number 8–10). The C-terminal domain is made up of strands from β7 to β11 that form a closely packed beta barrel structure. A double turn of α-helices is observed between β9 and β10 (KPTedELASK, residues number 100–102 and 105–109). So, as expected, *S. solfataricus* IF5A is a two-domain protein with an SH3-like N-terminal domain (residues 1–68) and an OB-Fold (69–131) C-terminal domain [45,46,47]. The InterPro study also confirms these folds.

As a second approach, in order to explore further the possible existence of different conformation of the protein in relation to its RNA substrates, we performed small-angle X-ray scattering (SAXS) measurements in the absence and in the presence of RNA. SAXS was useful to probe the molecular shape of *S. solfataricus* N-His-aIF5A directly in solution and exploited the possibility to carry out measurements at high temperature to obtain structural details about this highly thermophilic protein in a “close to physiological” state.

As described in Materials and Methods, all curves were recorded at Diamond Light Source (Didcot, UK), using the unmodified protein expressed in *E. coli*, with a 6 x Histidine flag at the N-terminal position. It was used only this recombinant protein because for this type of analysis a large quantity of protein is required (in total, 1.5 mg of protein), an amount difficult to obtain from that expressed in *Sulfolobus*, due to the difficulties in growing large volumes of culture at high temperature. Experimental SAXS curves of the translation factor aIF5A, measured at different protein concentrations and temperatures, are shown in Figure 3C (60 mM KCl) and Figure 3D (1000 mM KCl). Each frame of the two panels reports curves at the same temperature and at different protein concentrations (the color palette from blue to red refers to protein concentration from 0.5 to 10 mg/mL). The trend of the SAXS curves shows, at low q, the presence of large aggregates in combination with the presence of smaller particles justifying the Guinier/Porod analysis described in Appendix A. All SAXS data are shown in the form of Kratky plots ($q^{2}d\Sigma /d\Omega \left(q\right)$ vs. q) in Figure 3E and Appendix A. This representation gives information on the degree of compactness of the proteins: the bell-shaped behavior indicates that aIF5A maintains a compact structure for all the investigated conditions of temperature, protein concentration and KCl concentration. To note, the protein maintains a compact structure even at 63 °C, confirming its thermophilic characteristics. Experimental SAXS data are also shown in the form of Guinier plots (logarithm of dΣ/dΩ(q) vs $q^{2}$) in Figure 3F and Appendix A, together with the best fit obtained with the combined Guinier/Porod analysis, shown as a continuous black line. Dashed lines represent the contribution of the Guinier law (the first term of Appendix A), which agrees well with the linear trend of the experimental points. Thanks to this approach, the apparent aIF5A radius of gyration, $R_{g}$, in each measured condition, has been calculated. Results are shown in Appendix A, panels A and B: the radius of gyration ranges between 24 and 30 Å, confirming the presence of stable compact protein states, in agreement with the behavior of Kratky plots. Indeed, considering the experimental uncertainty, the value of $R_{g}$ does not seem to be strongly influenced by temperature, protein concentration and KCl concentration. From the fitted values of $K_{G}$ (Appendix A), the apparent aggregation numbers of the small particles in solution, $N_{agg,sml\;}$), has been determined. Results are reported as a function of temperature in Appendix A, panels C and D. It should be noticed that this number ranges between 1 and 2 in all experimental conditions, without any significant effect of the temperature, whereas protein concentration induces a slightly increase in the aggregation number. These achievements, obtained by using the asymptotic Guinier/Porod trend of SAXS curves, give a first indication that the smaller particles of aIF5A could be both monomers or dimers or, more likely, monomers that interact with a short-range attractive force. Based on the preliminary Guinier/Porod analysis, applied only to the low q region of the SAXS curves, the whole set of 82 SAXS curves has been fully analyzed, in whole the q range, with the GENFIT software, according to the model described in Appendix A, which describes a combination of Porod particles and interacting aIF5A monomers. Solid black lines in Figure 3C,D are the best fitting obtained with this method. To note, the best fitting curves have been plotted in a wider q range. The high quality of the fitting can be easily appreciated. We turn now to discuss the most relevant structural information derived from this advanced analysis. The trend of the aggregation number for what concerns the largest particles present in the sample, $N_{agg,por}$, (see Appendix A), is shown in Appendix A, panels E and F. Results clearly indicate that $N_{agg,por}$ decreases with temperature and increases with protein concentration, whereas it does not show any significant trend with KCl concentration. aIF5A is a thermophilic protein that performs its function at high temperatures. The fact that the number of aggregates decreases with increasing temperature could be related to the activation of the functions of the protein, which is less inclined to interact with other particles. From the fitted values of the fraction of all the aIF5A monomers aggregated in large particles, $x_{por}$, shown in Appendix A, panels G and H, is it straightforward to calculate the molar concentration of the aggregate particles, $C_{por=}\;x_{por}c{{}^{\circ}}_{5A}d_{wat}/\left(M_{5A}N_{agg,por}\;\right)$. Results are reported in the panel I and J of Appendix A. To note, these values are in the order of some tens of nM and appear to be quite constant with temperature. Considering the decreasing of $N_{agg,por}$ with temperatures, we can state that by increasing T, the aggregated particles just reduce their size, but their number remains quite constant. By considering a simple protein aggregation equilibrium process, $N_{agg,por}\left(5A\right)⇌{\left(5A\right)}_{N_{agg,por}}$, from the obtained results, we can calculate the apparent equilibrium constant of the aggregation process,
(1)$$K_{agg}=\frac{C_{por}\;}{C_{1}^{N_{agg,por}}}=e^{-\Delta G_{por}^{0}/RT}$$
where $C_{1}$=$\;\left(1-x_{por}\right)c_{5A}^{\circ }d_{wat}/M_{5A}\;$ is the molar concentration of 5A that remains in monomeric form. As a consequence, the change in the Gibbs free energy, $\Delta G_{por}^{0}$, can be derived from fitting parameters. In Appendix A, panels K and L, the ratio $\Delta G_{por}^{0}$/$N_{agg,por}$,which represents, on average, the contribution in Gibbs free energy provided by each 5A monomer forming the Porod aggregate, is reported. Values are quite independent on T and increase with protein concentration, varying approximatively from −20 kJ/mol at $c_{5A}^{\circ }=$ 0.5 mg/mL to −16 kJ/mol at $c_{5A}^{\circ }=$ 10 mg/mL. We discuss now the results regarding the interacting monomers, investigated, according to the second term of Appendix A, as a mixture of N = 10 different conformers, as shown in Appendix A. Values of the relative weights $w_{j}$ obtained because of the fitting parameters, according to Appendix A, are reported in Appendix A, panels M and N. Their values are very close to 1/N, with an almost negligible dependency on temperature, indicating that all the conformers found by QUAFIT contribute in a very similar manner to describe the heterogeneity of the aIF5A in the monomeric state. The monomer–monomer interactions are described by the HSDY potential, which has been optimized by fixing the ionic strength of the solution to the value known from the composition and the net number of elementary electric charge in a monomeric monomer to Z  = −4.34, calculated on the basis of the primary structure of aIF5A, as discussed in the previous section. The only fitting parameter of the HSDY potential are the ones describing the supposed linear variation with T and $C{{}^{\circ}}_{KCl}$ of the depth J of the attractive term at contact and the range d of its exponential decay. Plot of the linear trends of J, as a function of temperature, are reported in Appendix A, panels O and P, whereas the trends for d are shown in the panels Q and R of the same figure. Results deserve a comment. Firstly, both parameters, J and d, are almost constant with temperature, in agreement with the HSDY theory, which does not suppose a temperature variation of them. Concerning J, we can see that it changes from approximatively 17.5 kJ/mol at 60 mM KCl to 18.5 kJ/mol at 1000 mM KCl, indicating that the salt makes the contact between two monomers more sticky. It is worth to notice that the values of -J, which represent the energy stability at the contact, are very close to $\Delta G_{por}^{0}$/$N_{agg,por}$ indicating that the same energetics scenario describes both long range monomer–monomer interaction and monomer aggregation. To the sake of completeness, we report in Appendix A the structure factors S(q) (solid lines) and the effective structure factors $S_{M}\left(q\right)\;$(dotted lines), according to the best fitting parameters obtained by Genfit. The extrapolation at q = 0 of S(q) always falls between 1 and 2, a value that can be considered the apparent aggregation number of the network of aIF5A monomers, in fully agreement with results of Guinier/Porod analysis. By the Fourier transform of S(q), we have finally determined the radial correlation functions g(r), which describe, in the space of the distance rbetween two aIF5A monomers, the degree of correlation. Results are shown in Appendix A. The contact distance r= 2R, at which g(r) falls to zero, is clearly observable. In all cases, curves show a broad peak that extends from approximatively r = 36 Å up to 90 Å with high correlation g(r)≥1, indicating the radial region around one 5A monomer where it is likely to find a second monomer. In the presence of 1000 mM KCl (Appendix A), the broad peak is, in general, higher than for samples at 60 mM KCl (Appendix A), in agreement with the larger values of J.

### 3.4. SAXS Analysis of aIF5A in the Presence of RNA Substrate

In order to shed light on the role of rRNA in modifying the conformational/aggregational behavior of aIF5A, we performed SAXS experiments on samples with different aIF5A and rRNA concentrations. These experiments were performed at 63 °C. SAXS curves and the respective Kratky and Guinier plots are shown in Figure 4A–C, respectively. The log–log curves (Figure 4A) show the presence, at low *q*, of an upward curvature, indicating the presence of large particle, followed by a band at intermediate *q*, which changes with sample composition, most likely related to smaller particle. All Kratky plots (Figure 4B) show well defined bell shapes, indicating the presence of compact structures. On the basis of these qualitative observations, the SAXS curves were analyzed with a Guinier/Porod approach shown in Appendix A, which allows us to derive the average radius of gyration of all the smallest particles in solution, being the contribution of larger particles considered by the Porod law. As reported in Figure 4D, the radius of gyration of the samples with 0, 0.5 or 1 mg/mL of protein and 2.5 mg/mL rRNA (blue, green and yellow curves, respectively) are as large as 40 Å, whereas by increasing the protein concentration to 4 and 6 mg/mL and concomitantly decreasing rRNA concentration from 1.75 to 1 mg/mL (orange and red curves), $R_{g}$ decreases to 36 and 33 Å, respectively. In a simple approximation, by considering only, as small particles, the protein and small fragments of rRNA, the square of the average radius of gyration can be approximated by
(2)$$R_{g}^{2}=\frac{(c_{5A}/c_{\mathrm{rRNA}})M_{5A}^{-1}{\left(\nu_{5A}\Delta \rho_{5A}\right)}^{2}R_{g,5A}^{2}+w_{\mathrm{rRNA}}M_{\mathrm{rRNA}}^{-1}{\left(\nu_{\mathrm{rRNA}}\Delta \rho_{\mathrm{rRNA}}\right)}^{2}R_{g,\mathrm{rRNA}}^{2}}{(c_{5A}/c_{\mathrm{rRNA}})M_{5A}^{-1}{\left(\nu_{5A}\Delta \rho_{5A}\right)}^{2}+w_{\mathrm{rRNA}}M_{\mathrm{rRNA}}^{-1}{\left(\nu_{\mathrm{rRNA}}\Delta \rho_{\mathrm{rRNA}}\right)}^{2}}$$
where $c_{j}$, $M_{j}$, $\nu_{j}$, $\Delta \rho_{j}$ and $R_{g,j}$ are the *w/v* concentration, the molecular weight, the electron density contrast and the radius of gyration of the *j*-species, with *j* standing for aIF5A or rRNA. In this equation, $w_{\mathrm{rRNA}}$ is the fraction of rRNA mass involved in small particles. We have considered canonical values of the contrasts for proteins and RNA ($\Delta \rho_{5A}=$0.09 e/Å^3^ and $\;\Delta \rho_{\mathrm{rRNA}}\;$ = 0.21 e/Å^3^) and we have fixed $M_{\mathrm{rRNA}}\;$ = 39 kDa, corresponding to the molecular weight of 5S rRNA. Hence, the values of $R_{g}$ obtained, for different ratios $c_{5A}/c_{\mathrm{rRNA}}$, by the Guinier/Porod analysis shown in Figure 4C can be fitted with Equation (2) in order to obtain the unknown parameters $w_{\mathrm{rRNA}}$, $R_{g,5A}$ and $R_{g,\mathrm{rRNA}}$. Best fit results are reported in Figure 4D, with optimum parameters $w_{\mathrm{rRNA}}\;$ = 0.6 ± 0.3, $R_{g,5A}\;$ = 27 ± 3 Å (similar to the values obtained by the GENFIT analysis of SAXS data of proteins, see Appendix A, panel A) and $R_{g,\mathrm{rRNA}}\;$ = 40.6 ± 0.6 Å.

### 3.5. Structural and Functional Features of S. solfataricus aIF5A by InterPro Analyses

To elucidate the potential mechanisms of RNA binding/cleavage of *S. solfataricus* aIF5A that is a two-domain protein with an SH3-like N-terminal domain (residues 1–68) and an OB-Fold (69–131) C-terminal domain (Figure 5A) [45,46,47] we have analyzed different structural properties of the protein. The SH3 module consists of approximately 50 amino acid residues and this represents a peculiar fold composed by 5/6 β-strands organized in two anti-parallel β-sheets. The SH3 domain is very often engaged in contacting/binding proline-rich motifs [48]. The OB-fold, on the other hand, has 5/6 β-strands made by 60–70 amino-acid residues. A typical OB-fold domain is made by two β-sheets linked together by extensive loops. In the loop between the two β-sheets, it is also possible to find an α-helix. This fold’s function is limited to binding single/double-stranded DNA/RNA (i.e., dsDNA, dsRNA, ssDNA, ssRNA) or sugar [46]. The presence of aromatic and basic residues is an important feature of the OB-fold. The aromatic residues are engaged in stacking interactions with the nucleotides, whereas the basic residues stabilize the phosphate moieties on the backbone of the nucleic acids [46,49]. Therefore, we highlighted the key basic and aromatic residues on the *S. solfataricus* aIF5A OB-fold structure (Figure 5A) [50]. We also found that the *S. solfataricus* aIF5A C-terminal OB-fold arranges several aromatic and basic amino acid residues (i.e., K72, H73, K109, K111, W119, R124, R125, K126, R129 and K131). Another site where critical amino acids are located is at the interface between the two domains of *S. solfataricus* aIF5A (i.e., K12, Y16, R25, F50 and K54). Interestingly, a pairwise alignment (Appendix A) with the archaea *Halobacterium* sp. NRC-1 aIF5A sequence revealed that several of the previously identified residues whose substitution causes a reduction in the RNAse activity (i.e., K72, H73, R125, K126) are conserved (Wagner et al. [25]). Furthermore, the analyses of the structural models together with the multiple alignment (Appendix A) have allowed us to highlight two amino acids that are probably also involved in the RNAse activity of *S. solfataricus* aIF5A (i.e., G9 and E120) (Appendix A). Therefore, we hypothesize that all the amino acids listed so far mediate the interaction between *S. solfataricus* aIF5A and the RNA, confirming what was already reported by Wagner and colleagues [25], and they could have also a crucial role in the catalytic activity of the protein. We aligned twenty sequences from various archaeal species to better understand the functional role of these residues, and the majority of them are conserved (Appendix A). For a broader comparative structural analysis, we have retrieved all of the IF5A proteins available in the PDB from both eukaryotic and archaeal organisms. Additionally, we used SwissModel to obtain the model of the *Halobacterium* sp. NRC-1 IF5A protein structure (i.e., SM-Hsa-IF5A). Because model accuracy impacts the physico-chemical properties of the protein structure, we also utilized AlphaFold2 [51] to predict the structure of *S. solfataricus* aIF5A. (i.e., AF-Sso-IF5A). The AF-Sso-IF5A model was predicted with high confidence (Appendix A), with an RMSD of 0.843 Å when superimposed on the SM-Sso-IF5A structure (Appendix A). We have collected 19 structures (including the SM-Sso-IF5A and the AF-Sso-IF5A): seven from Archaea and twelve from Eukarya (Appendix A). The superimposition of all structures relative to *S. solfataricus* aIF5A clearly shows that these proteins are well conserved in both eukaryotic and archaeal organisms (Appendix A). To further understand the nature of the *S. solfataricus* aIF5A electrostatic contribution to RNA binding, we have also reported the electrostatic potential surface (EPS) calculated for many IF5A structures estimated using the Adaptive Poisson–Boltzmann Solver (APBS) [42]. The hypothetic binding surface for the RNA was located in the grove produced in the hinge region based on the *S. solfataricus* aIF5A EPS (Figure 5B,C), in accordance with the previously identified residues. This surface matches with the OB fold in which an oligonucleotide binding cleft is present [52]. We observed that the only protein with an evidently positive potential in the C-terminal region was the one from *S. solfataricus* (Figure 5B,C and Appendix A). Furthermore, only a few Archaea structures (from *Pyrobaculum aerophilum* and *Pyrococcus horikoshii*) exhibit this specific chemico-physical feature, albeit less prominently. All the other eukaryotic structures have a well-defined negative C-terminal domain (Figure 5C). Because of these results, we may hypothesize that *S. solfataricus* aIF5A can consistently bind more strictly RNA. Indeed, an extended positive electrostatic potential between strands β2-β3 and β10-β11 forms the binding surface where an induced-fit oligo-binding process with broad specificity may be engaged. The *S. solfataricus* aIF5A C-terminal retains the documented features of the OB-fold in both the location of its oligonucleotide-binding interface and the presence of exposed aromatic (i.e., Y16, F50, W119) and positively charged sidechains (i.e., K12, R25, K54, K72, H73, K109, K111, R124, R125, K126, R129 and K131) [53,54].

## 4. Discussion

The present work reports a structural and functional characterization of one of the most abundant and conserved translation factors, the aIF5A from *S. solfataricus*. The functions and properties of this protein have been studied in Eukarya and in Bacteria but remain somewhat elusive in the archaeal kingdom. In the hyperthermophilic model *S. solfataricus*, we show that this protein is always expressed during growth and that the bulk of aIF5A was found in the free form and was not ribosome associated [14], suggesting that it may serve other functions, beyond its conserved role as a translation factor. In this work, we provide some hints supporting the additional endonuclease activity of aIF5A from *S. solfataricus*.

First, we demonstrate that in *S. solfataricus* aIF5A is an RNA-binding protein in vivo, showing by RIP experiments, that it is associated with different types of RNA including two mRNAs (2184 and 0910) and one ncRNA (ncRNA98).

In addition, aIF5A is endowed with an endonuclease activity. We performed several in vitro degradation assays and showed that aIF5A can cleave the two mRNAs previously identified (2184 and 0910) as being aIF5A-associated in vivo, as well as rRNA 16S and 23S isolated from 30S and 50S ribosomal subunits. On the contrary, the ncRNA98 as well as a mixture of small RNAs, mainly containing tRNAs, are resistant to aIF5A degradation activity.

Our results are in agreement with previous findings, where we have hypothesized that cleavage was more likely to occur in single-stranded areas or bulges. Here, we show for the first time a clear preference of aIF5A for long RNA molecules such as rRNAs or mRNAs, that fold in more complex secondary structures, thus exposing more single-strand regions in loops as compared to shorter RNA, as tRNAs or ncRNAs, structurally more compact.

In all cases, hypusine seems to be irrelevant for the RNase activity, even though on one of the mRNAs analyzed, the hypusinated protein seems to be slightly more efficient in the cleaving activity. This may indicate that different RNAs may interact in different ways with the protein, and therefore, in some complexes, the presence of hypusine may favor the enzymatic activity.

Previous findings in yeast proposed that eIF5A occurs in solution as a dimer, and that dimerization is RNA-dependent [26,31]. Additional indications come from other archaeal species: the activity of *Halobacterium* aIF5A is dependent on its oligomeric form and it probably forms dimers [25], while crystallographic investigation in *M. jannashii* shows that it exists as a dimer in certain crystals [30]. Based on this information, we decided to analyze the *S. solfataricus* protein in vivo, investigating the influence of the RNA molecules might have on its conformation (i.e., monomeric or dimeric). To this end, we fractionated a control cell lysate and a cell lysate incubated with RNase A on glycerol gradients, and we explored the distribution of the native protein in the two samples. The results show that in the control lysate, aIF5A appears to migrate as a monomer, but in a lysate treated with RNase A, aIF5A shifts its position migrating in higher molecular weight fractions (around 34–40 kDa). This result indicates that in the absence of RNA, the protein may form either dimers or higher molecular weight complexes associating with other proteins. Therefore, the protein is a monomer in vivo most likely by its association with RNA molecules, and these RNAs can either be its binding partners or fragments deriving from its RNA-degrading activity. If RNAs are completely removed, the protein tends to aggregate, forming dimers or associating with other proteins.

SAXS experiments were performed to gain more information on the solution structure of aIF5A and on its tendency to dimerize. The results have shown, in addition to a strong stability of the protein, which is not influenced by variations of the different parameters analyzed (temperatures, protein concentration, KCl concentration), that the main conformation adopted in solution by N-His-aIF5A is monomeric, with a slight tendency to form oligomers. This can be explained by the fact that the recombinant protein purified from *E. coli* is also associated in a non-specific manner to bacterial RNAs (data not shown) [26,31]. In addition, SAXS analyses were performed to obtain a structural characterization of the protein in the presence of RNA molecules at 63 °C. Mixtures of N-His-aIF5A and rRNA were analyzed and the results, elaborated and presented in Figure 4D, show a decreasing trend of the Rg as a function of the increasing protein:RNA ratio. rRNA is a substrate for the protein-cleaving activity, and therefore, we may hypothesize that increasing the amount of protein leads to degradation of most of the RNA molecules. The Rg we observe at the highest ratio, which approaches values close to those observed for the monomer, may correspond to the protein in the monomeric form, probably associated with short RNA fragments.

Finally, we analyzed the structural features of aIF5A by computational approaches, pinpointing amino acids that could be important for interaction and for cleavage of RNAs. Several aromatic and basic amino acid are localized in the OB-fold or at the interface between the two domains. These residues could be important for binding RNA nucleotides through stacking interactions (aromatics), or through stabilization of the phosphate moieties on the nucleic acid backbone (basics) (Figure 5A). Concerning the amino acid involved in the catalytic activity, their conservation between *Halobacterium* and *S. solfataricus* suggests a possible similar catalytic mechanism. To better understand the interaction mechanism of the *S. solfataricus* aIF5A with RNA, the electrostatic potential surface (EPS) was calculated for several IF5A structures. The protein from *S. solfataricus* was the only one having a prominent positive potential in the C-terminal region (Figure 5B,C), and this could be the reason why *S. solfataricus* aIF5A may have a strong tendency to bind and degrade RNAs.

Overall, the results seem to indicate that in the cell, a fraction of aIF5A is bound to translate ribosomes, while the rest is in the cytoplasm, most of it associated with RNA molecules. Some of these molecules (e.g., the short ncRNA98) are binding partners, and it remains to be clarified whether the association with aIF5A has a functional meaning, and whether other RNA (mRNAs) are substrates for aIF5A degradation activity. Further studies are necessary to clarify in more detail the mechanism of cleavage of Sso aIF5A and to investigate the role of the protein in RNA-processing pathways in *S. solfataricus*. In addition, information on mechanisms by which RNA species reach their mature forms or are degraded are still fragmentary in Archaea. Several endonucleases have been identified from the information gathered from Bacteria and Eukarya, as the orthologue of RNase P, the splicing endonuclease End A, aRNase Z, aPelota, members of the ß-CASP ribonuclease family (aCPSF1, aCPSF2), aNob1 and CRISP Cas6 [55]. In this framework, aIF5A could be a new RNase candidate that, in complex with other components, participates in RNA processing and decay paths in *S. solfataricus*.

## 5. Conclusions

In conclusion, we have shown that the protein exists mainly as a monomer associated with RNA molecules and probably remains RNA bound even when it acts as a ribonuclease. Indeed, being an endonuclease with preference for single strand loops, it is conceivable to assume that the products of the enzymatic activity are RNA fragments which may remain bound to the protein, helping to maintain its monomeric conformation.

The different functions of aIF5A (i.e., RNA binding and RNA degradation) are interconnected, and this “constitutive” association with RNA could be functional in providing a constant pool of monomeric protein ready to associate, when needed, with stalling ribosomes.

## Figures and Tables

**Figure 1 biomolecules-12-01432-f001:**
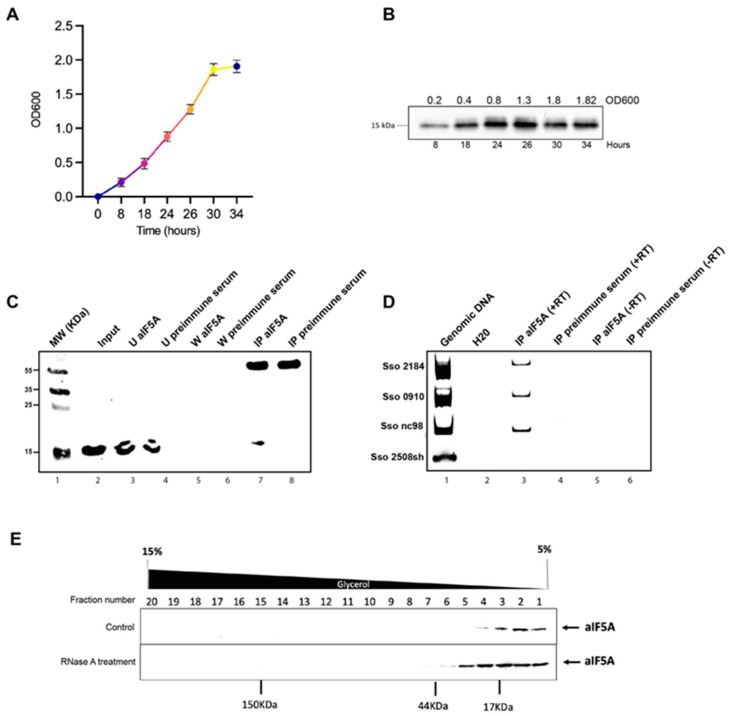
Expression of *S. solfataricus* aIF5A and its interaction with RNA molecules in vivo: (**A**) Growth curve of *S. solfataricus* P2 in Brock medium supplemented with 0.2% NZ amine and 0.2% sucrose. Growth of three replicates was further monitored. An amount of 15 mL of culture at 0.2 OD (8 h), 0.4 OD (18 h), 0.8 OD (24 h), 1.3 OD (26 h), 1.8 OD (30 h) and 1.82 (34 h) were harvested; (**B**) 20 μg of total protein from each lysate at different densities during the cell cycle were subjected to SDS-PAGE electrophoresis followed by Western blot and probed with anti-aIF5A; (**C**) native aIF5A immunoprecipitation from *S. solfataricus* post-ribosomal fraction (S100) with anti-aIF5A antibody and pre-immune serum and analysis of immunocomplexes by Western blotting, using anti-aIF5A. Lane 1: molecular weight protein marker; lane 2: input of native aIF5A in S100; lane 3–4: unbound fractions; lane 5–6: last wash fractions; lane 7–8: elution fractions.; (**D**) Reverse-Transcriptase-PCR with primers that amplified 2184, 0910, 2508sh mRNA and ncRNA98. Lane 1: *S. solfataricus* P2 genomic DNA amplification (control for PCR); lane 2: sample with H20, to exclude any contamination of genomic DNA in water used for PCR; lane 3–4: RNA-coimmunoprecipitated with native aIF5A (IP 5A); lane 5–6: RNAs in control samples of immunoprecipitation with preimmune serum. The reactions were carried out with (+RT) and without (-RT) Reverse Transcriptase, to exclude any DNA contamination; (**E**) Fractionation of 0.5 mg (total proteins) *S. solfataricus* S30 extract on glycerol gradients 5–15%, under normal condition or with a previous treatment of the extract with the RNase A. Aliquots of each fraction were resolved by SDS–polyacrylamide gel electrophoresis and immunoblotted with anti-aIF5A antibody.

**Figure 2 biomolecules-12-01432-f002:**
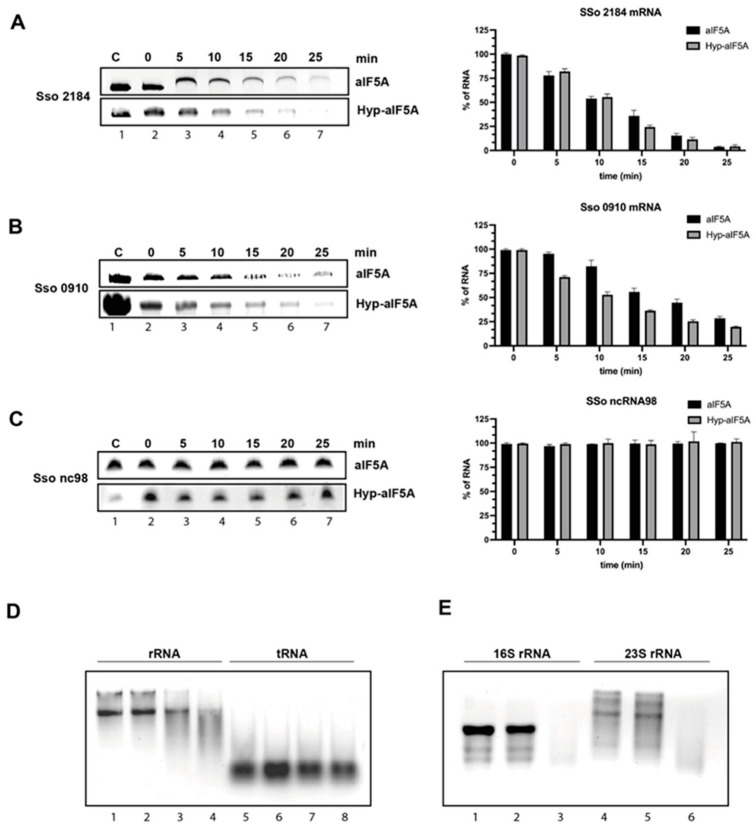
aIF5A shows differences in its ribonucleolytic activity based on different RNA classes: Degradation assay of *S. solfataricus* 2184 mRNA (**A**) 0910 mRNA (**B**) or ncRNA 98 (**C**) with recombinant N-His-aIF5A (produced in *E. coli*, without hypusination) and Hyp-aIF5A (produced in *S. solfataricus* and hypusinated). Lane 1, RNAs incubated for 25 min at 65 °C in absence of proteins; lanes 2–7, time-course of degradation of RNAs with N-His-aIF5A (upper panel) and Hyp-aIF5A (lower panel) for 25 min at 65 °C. ImageLab software were used to quantify the RNA and the signal of the 0 min sample was set to 100%. The graphical representation shows the average of three independent experiments; the error bars represent standard deviations. (**D**) Degradation assay of total rRNA and tRNA with recombinant N-His-aIF5A. Total rRNA (lane 1) or total tRNA (lane 5) without incubation and in the absence of protein; total rRNA (lane 2) or total tRNA (lane 6) incubated for 30 min at 65 °C in the absence of protein; total rRNA (lane 3) or total tRNA (lane 7) incubated for 30 min at 65 °C in presence of 100 pmol (lane 3 and 7) or 200 pmol (lane 4 and 8) of N-His-aIF5A. (**E**) Degradation assay of 23S and 16S rRNA with recombinant N-His-aIF5A. 23S (lane 1) and 16S (lane 4) without incubation and in the absence of protein; 23S rRNA (lane 2) or 16S tRNA (lane 5) incubated for 30 min at 65 °C in the absence of protein; 23S rRNA (lane 3) or total tRNA (lane 6) incubated for 30 min at 65 °C in presence of 100 pmol of N-His-aIF5A.

**Figure 3 biomolecules-12-01432-f003:**
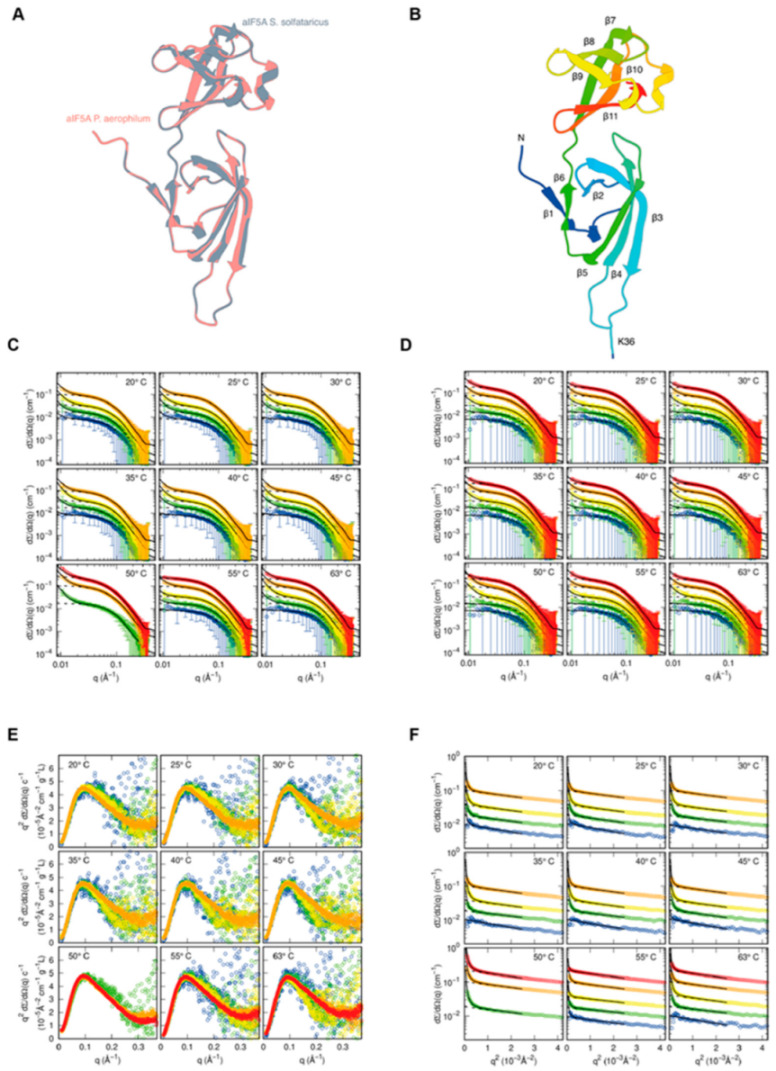
Structural characterization of *S. solfataricus* aIF5A in solution: (**A**) The structures of modeled *S. solfataricus* aIF5A superimposed on *P. arophilum* aIF5A (PDB ID 1BKB) are shown in gray and light coral, respectively. The RMSD estimated between the C-α atoms of the structures is 0.120 Å; (**B**) 3D topology of *S. solfataricus* aIF5A modeled structure. The molecular structure is colored from the N-terminus (dark blue) to the C-terminus (red). In the depiction the location of the lysine 36 that is normally hypusinated is highlighted; (**C**,**D**) SAXS curves shown in log–log plots of the N-His-aIF5A. Each frame reports data at the same temperature, as indicated. The protein concentration of each sample is: 0.5 mg/mL (blue); 1 mg/mL (green); 2 mg/mL (yellow); 5 mg/mL (orange); 10 mg/mL (red). Samples in panel **C** contain 60 mM KCl, and the ones in panel D contain 1000 mM KCl. Error bars are reported every 10 points, for clarity Solid black lines are the best fit obtained with GENFIT (Appendix A); (**E**) Kratky plots of data reported in panel **C**. Curve have been divided by the protein concentration; (**F**) Guinier plot of data reported in (**C**). The solid black lines in panel f are the best Guinier/Porod fits (Appendix A), whereas the dashed black lines are the Guinier law contribution (first term of Appendix A). Kratky and Guinier plots of data corresponding to panel D are shown in Appendix A.

**Figure 4 biomolecules-12-01432-f004:**
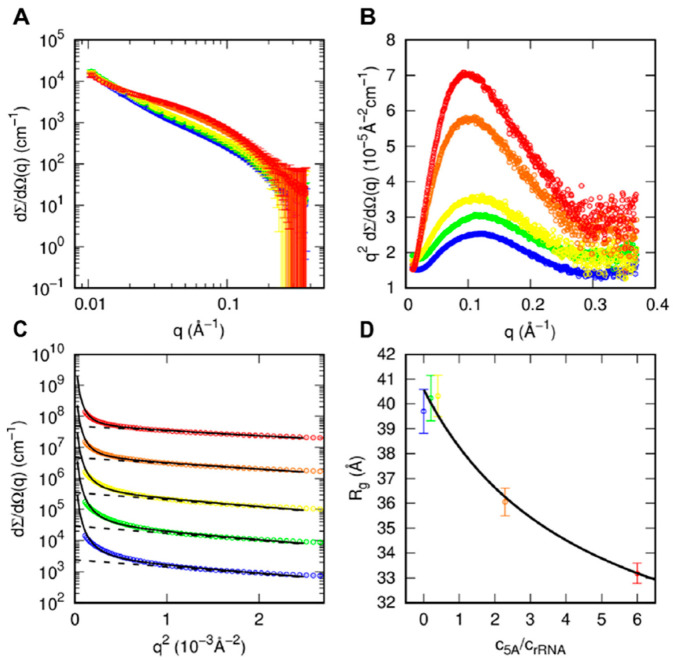
SAXS analysis of N-His-aIF5A in the presence of rRNA at 63 °C: (**A**–**C**) SAXS curves reported in log–log, Kratky and Guinier plots. The protein and the rRNA concentrations of each sample are: 0 and 2.5 mg/mL (blue); 0.5 and 2.5 mg/mL (green); 1 and 2.5 mg/mL (yellow); 4 and 1.75 mg/mL (orange); 6 and 1 mg/mL (red). These values correspond to the protein/rRNA *w*/*w* ratio ranging from 0 to 6. Curves in panel **C** have been multiplied by a factor 10 for the sake of a better visualization. The solid black lines in panel **C** are the best Guinier/Porod fits (Appendix A), whereas the dashed black lines are the Guinier law contribution (first term of Appendix A). (**D**) The obtained radii of gyration as a function of the protein/rRNA ratio. The solid black line is the best fit obtained with Equation (2).

**Figure 5 biomolecules-12-01432-f005:**
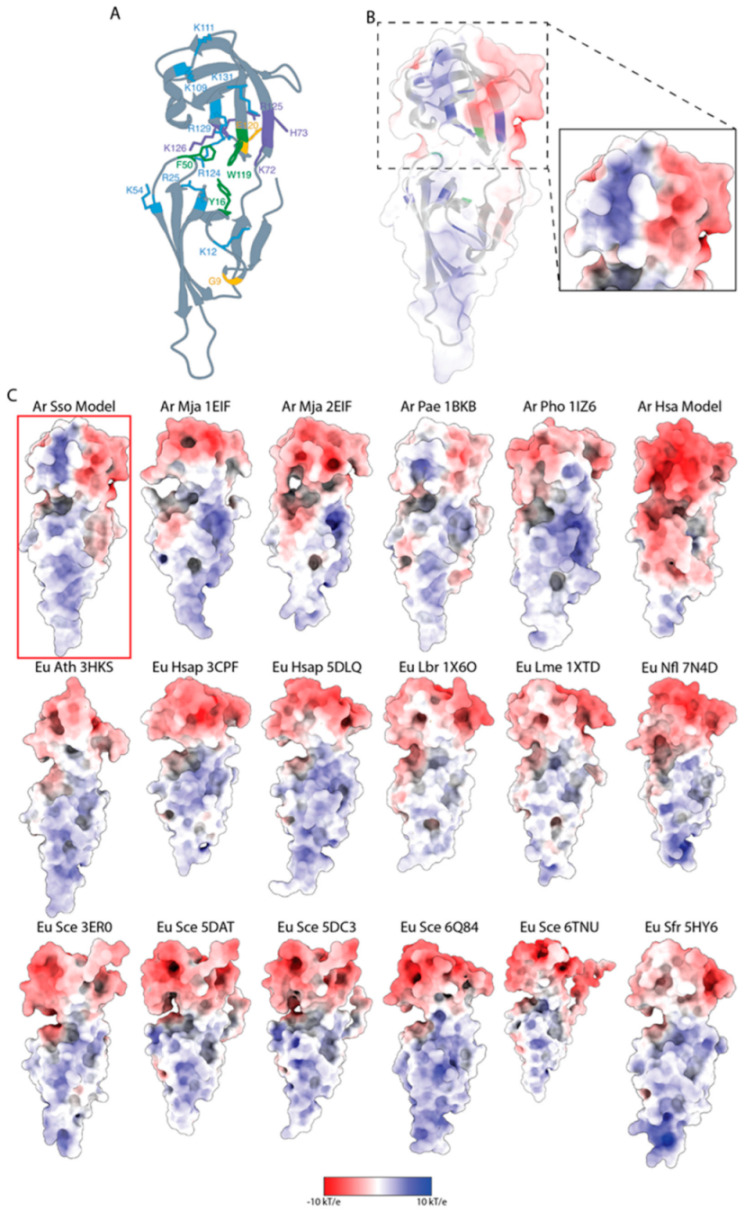
IF5A RNA-binding and cleavage structural characteristic. (**A**) 3D ribbon illustration of *S. solfataricus* aIF5A where basic residues (i.e., K12, R25, K54, K109, K111, R124, R129, and K131), aromatic residues (i.e., Y16, F50, W119), common residues (i.e., K72, H73, R125, and K126) identified from the pairwise alignment with *Halobacterium* sp. NRC-1 amino acid sequence (reported from the work by Wagner et al. [25]) and hypothetical residues (i.e., G9 and E120) involved in RNAse activity (reported from the work by Wagner et al. [25]) are depicted in light blue, green, purple and yellow, respectively. (**B**) *S. solfataricus* aIF5A 3D ribbon illustration with electrostatic potential mapped on molecular surface. The ribbon structure depicts the essential basic and aromatic residues in blue and green, respectively. The electrostatic potential projected on the molecular surface of the OB-Fold region of *S. solfataricus* aIF5A is shown in the inset. (**C**) Electrostatic potentials are represented by the color of the molecular surfaces of several archaea and eukaryotic IF5A proteins: red is negative, blue is positive, and white is neutral. The model structure of *S. solfataricus* aIF5A is outlined in red. All the structures were refined and utilized to determine the EPS with the ABPS tool applying continuum solvation techniques. All structures were superimposed on *S. solfataricus* aIF5A to achieve the same spatial orientation. Surface potentials range from −10.0 kT/e (red) to 10 kT/e (blue). When no PDB code is given, the structure is a model. Ar, Archaea; Eu, Eukarya; Sso, *Sulfolobus solfataricus; Mja, Methanocaldococcus jannaschii; Pae, Pyrobaculum aerophilum; Pho, Pyrococcus horikoshii; Hsa, Halobacterium sp.* NRC-1; *Ath, Arabidopsis thaliana; Hsap, Homo sapiens; Lbr, Leishmania braziliensis; Lme, Leishmania Mexicana; Nfl, Naegleria fowleri; Sce, Saccharomyces cerevisiae; Sfr, Spodoptera frugiperda*.

## Data Availability

Not applicable.

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
