# Peer review of "Structural–Functional Relationship of the Ribonucleolytic Activity of aIF5A from *Sulfolobus solfataricus"

_biomolecules, 2022, doi:10.3390/biom12101432_

Round 1

Reviewer 1 Report

The manuscript by Romagnoli et al. reports an interesting study of  translation factor aIF5A from the crenarchaeal S. solfataricus model organism. Most of aIF5A is found in a free form, not associated to the ribosome, supporting a function of the protein beyond that in translation. RNase activity had already been reported for another archaeal aIF5A. Accordingly, it is reported here that S. solfataricus aIF5A displays RNA-binding and hydrolytic properties, especially on long structured RNAs. This RNase activity does not depend on hypusination. Biochemical characterization suggests that the protein may dimerize but that  dimerization may be reversed by RNA. This conclusion is corroborated by SAXS experiments made with both the protein alone and in the presence of RNA. Finally, possible residues involved in RNAse activity are proposed on the basis of 3D structure modelling.

This is an interesting study that convincingly suggests that aIF5A may have an additional function, beyond its known one in translation. The experiments are thorough and adequately performed. The following remarks may help the authors to enhance the clarity of their manuscript.

1- The SAXS experiments made in the presence of RNA need several readings to be understood. Because the rRNA used contained both large and small fragments, it would be much helpful to explain at the beginning of section 3.4 (page 14) the rationale for the experiments and for the interpretation of the SAXS data.

2- Demonstration of an RNase activity is always difficult because the possible presence of contaminating activity. The authors should further discuss this point. Perhaps, the similar activities of the two aIF5A preparations (from E. coli and S. solfataricus) may be used by the authors to further support their conclusions.

3- Because Swiss-Model is homology based, it is not surprizing that SM-Sso-IF5A is similar to the eukaryal and archaeal counterpart (page 10, line 402). This should be mentionned. Moreover, the electrostatic potential surfaces (Figure 5) may somewhat vary depending on the local accuracy of the model. Thus, for Sso-IF5A, it may be worth calculating an AlphaFold 2 model.

Minor comments:

page 2, line 55: many pieces of evidence (rather than many evidences).

page 6, line 286: then instead of than.

Reviewer 2 Report

The authors report a structural and functional characterization of aIF5A from S. solfataricus. New experimental structural information in present paper was obtained with low resolution by SAXS and models from computation analysis looks identical to the crystal and solution structures of eIF5a homologues from PDB database and are not conceptually different from any of the previously published structures. The evidence for aIF5a dimerization is also weak. As an advice, I would recommend analyzing the dynamics of the 15N labeled protein by NMR spectroscopy to prove this statement.

Nevertheless in vivo and in vitro experiments and information about RNA-binding are new and will be of wide interest.

 It should be suitable for publication following revision, as outlined below:

1)     In the Introduction section line 30 authors mentioned that "all proteins undergo a post-translational modification", however not all bacteria have modification of EF-P. Moreover, only lysinylation is mentioned for Bacteria, although there are also rhamnosylation and 5-aminopentanylation modifications (please look at the review DOI: 10.1134/S0026893319040034). I recommend that this information be added in Introduction section. 

2)     In Figure 1A the error bars are missing

3)     The text indicates that aIF5a from S. solfataricus contained a 6-His tag, but does not mention that it was cleaved by proteases. If the His-tag was not cleaved, it should be stated that its presence does not affect the function of the protein.

4)     Please clarify why a such high concentration of KCL (1000 mM) and temperature of 63 degrees were chosen for SAXS? At high salt concentration protein could be not folded but present as a melting globula.

Reviewer 3 Report

Romagnoli et al. studied aIF5A from S. solfataricus to understand its structure and functions related to RNA metabolism. They found aIF5A have ribonuclease activity, with a specificity to long RNA. Also, they claimed that aIF5A is mainly monomer, and is preferably in monomer state when binding to RNA. In the end, by modeling the structure of aIF5A to homologues, they concluded a positive charge surface on aIF5A might be responsible for its RNA binding activity.

However, I am suspicious about most of these claims, without further evidence. For example, the authors claim that aIF5A has a ribonuclease activity, do they find the active site on this protein by comparing it with other known ribonuclease family proteins? While the assays are poorly controlled, without ruling out the possibility of RNase contamination from buffers or other sources (see detailed below). For oligomeric state of the protein, the authors used gradient centrifuge to show a shift of the peak upon RNase treatment, and they also didn’t rule out the possibility of binding to other factors. While the SAXS data is trying to prove the similar findings of the oligomer state, this is still an indirect readout, as the RNA binding might change the conformation of protein so that the “size” is changed. Thus, it’s recommended to use purified proteins or other techniques such as SEC-MALS to test the hypothesis. 

In summary, I wouldn’t consider the current manuscript ready for publication without further clear evidence to support these conclusions. 

Some minor points:

Figure 1B: controls are needed for western blot to quantify the expression level compared to cell growth. 
Figure 1E: the amount of proteins seems not the same between control and RNase A treatment, based on the band intensities. Moreover, the proteins stay too close to the gradient starting point, have you tried to use even lower glycerol concentration. I would recommend using purified recombinant aIF5A to test in this assay. 

Figure 2B and 2C: the bar graph didn’t match the gel, especially on the control sample. 

Would you comment why you would include His tag on different positions when expressing from E. coli or S. solfataricus?

Round 2

Reviewer 3 Report

I am sorry that I am still suspicious about the ribonuclease activity of IF5A in their claim. While it shares nucleic acids binding domains with other RNases, it doesn’t mean it’s a ribonuclease. While previous studies (https://doi.org/10.1074/jbc.M701166200) pointed out that archaeal IF5A has ribonuclease activity, which is boosted through hypusination. The authors didn’t see this in S. solfataricus. Importantly, this paper didn’t find the active site either. Do authors see similar product bands as seen in the above paper? In the end, I would like to see a control experiment of RNA decay assay with an IF5A mutant, it could be a mutant that blocks oligomer formation or RNA binding mutant, if active sites could not be characterized. Thanks!

In terms of Fig2, the authors should also include the timepoint of control samples throughout this 25min experiments to demonstrate the control samples not degrading. Although the authors responded that the bar graph was normalized to the 0’ time point, but the legends still indicate it was normalized to the control sample. So please make a time-point series for the control sample (no protein) as well. 

A minor issue about Fig2, the raw images from Fig2E looks over-exposed or overloaded.
